# Urine dicarboxylic acids change in pre-symptomatic Alzheimer's disease and reflect loss of energy capacity and hippocampal volume

K. J. Castor[1], S. Shenoi[1], S. P. Edminster[1], T. Tran[2], K. S. King[2], H. Chui[3], J. M. Pogoda[4], A. N. Fonteh[1]*, M. G. Harrington[1]*

1 Neurosciences, Huntington Medical Research Institutes, Pasadena, CA, United States of America, 2 Clinical MR Unit, Huntington Medical Research Institutes, Pasadena, CA, United States of America, 3 Department of Neurology, Keck School of Medicine, University of Southern California, Los Angeles, CA, United States of America, 4 Cipher Biostatistics & Reporting, Reno, NV, United States of America

* alfred.fonteh@hmri.org (ANF); michael.harrington@hmri.org (MGH)

**Data Availability Statement:** All relevant data are within the manuscript and its Supporting Information files.

## Abstract

Non-invasive biomarkers will enable widespread screening and early diagnosis of Alzheimer's disease (AD). We hypothesized that the considerable loss of brain tissue in AD will result in detection of brain lipid components in urine, and that these will change in concert with CSF and brain biomarkers of AD. We examined urine dicarboxylic acids (DCA) of carbon length 3–10 to reflect products of oxidative damage and energy generation or balance that may account for changes in brain function in AD. Mean C4-C5 DCAs were lower and mean C7-C10 DCAs were higher in the urine from AD compared to cognitively healthy (CH) individuals. Moreover, mean C4-C5 DCAs were lower and mean C7-C9 were higher in urine from CH individuals with abnormal compared to normal CSF amyloid and Tau levels; i.e., the apparent urine changes in AD also appeared to be present in CH individuals that have CSF risk factors of early AD pathology. In examining the relationship between urine DCAs and AD biomarkers, we found short chain DCAs positively correlated with CSF $A\beta_{42}$, while C7-C10 DCAs negatively correlated with CSF $A\beta_{42}$ and positively correlated with CSF Tau levels. Furthermore, we found a negative correlation of C7-C10 DCAs with hippocampal volume (p < 0.01), which was not found in the occipital volume. Urine measures of DCAs have an 82% ability to predict cognitively healthy participants with normal CSF amyloid/Tau. These data suggest that urine measures of increased lipoxidation and dysfunctional energy balance reflect early AD pathology from brain and CSF biomarkers. Measures of urine DCAs may contribute to personalized healthcare by indicating AD pathology and may be utilized to explore population wellness or monitor the efficacy of therapies in clinical trials.

**Funding:** JMP is employed by and receives salary from Cipher Biostatistics & Reporting. The funder had no role in study design, data collection and analysis, decision to publish, or preparation of the manuscript.

**Competing interests:** JMP is employed by and receives salary from Cipher Biostatistics & Reporting. The analysis by our consultant (JMP) from Cipher Biostatistics & Reporting does not alter our adherence to PLOS ONE policies on sharing data and materials.

## Introduction

Alzheimer's disease (AD) is the most common form of dementia, the sixth leading cause of death in the US, and the fourth leading cause of death in African Americans [1]. AD is characterized by extracellular β-amyloid deposition in the brain [2], followed by intracellular neurofibrillary tangles of hyperphosphorylated Tau proteins, accompanied by neuronal loss [3]. All attempts to reduce amyloid deposition in dementia have been unsuccessful in preventing or slowing neurodegeneration [4] and cognitive function, thus efforts are now focused on treatment at earlier stages of pathology [5]. However, methods to select patients with early AD pathology are limited by incomplete understanding of early pathophysiology and lack of biomarkers to predict the onset of AD in a cognitively healthy (CH) individual. Aims to improve this selection process [6] include clinical trials in mutation carriers with autosomal dominant AD, whose estimated clinical onset is more reliable based on each person's family history [7]. This early onset disorder is rare and pathologically distinct from sporadic AD, for which the lack of non-invasive, widely usable, predictive biomarkers is a substantial bottleneck for properly designing trials in individuals prior to symptom onset.

The principal validated biomarkers for AD rely heavily on molecular changes in the known amyloid/Tau pathology of AD, represented by decreased β-amyloid and increased Tau in cerebrospinal fluid (CSF) [8–10], and/or increased brain amyloid or Tau by positron emission tomography (PET) [11]. These techniques are not widely available or applicable to many patients due to the invasiveness of CSF collection and PET imaging and the high expenses for these procedures; furthermore, although useful to distinguish clinical groups, the margin of error for predicting onset of clinical deterioration may be 10–20 years [12–14]. Other candidate biomarkers from invasive studies include other CSF proteins [15–17], blood measures of Tau or amyloid [18], metabolites [19], or exosomes [20]; and, from non-invasive urine collection, proteins [21], and neural thread protein [22, 23]. Recent reports [24, 25] indicate great promise for AD blood biomarkers (amyloid and others), but these preliminary candidates remain to be further validated, and the need for more predictive molecular biomarkers is still widely recognized [26].

Amongst the many fluids that the body produces, we have focused our biomarker discovery efforts on urine samples, since it is a rich source of molecules that can be quantified to reflect brain and body function. Of the several metabolites found in urine, we targeted dicarboxylic acids (DCA) because they are implicated in several processes associated with AD pathology. For example, DCAs are formed from the oxidative breakdown of unsaturated fatty acids [27, 28] and the increase in oxidative stress associated with AD is predicted to alter DCA formation from long chain monounsaturated and polyunsaturated fatty acids. Several DCAs such as succinic acid and glutaric acid contribute to energy metabolism and changes in their levels may impact mitochondrial function. Mitochondrial function and energy imbalance are proposed to contribute to AD pathology. DCAs are known to inhibit mitochondrial ATP production and alter respiration. Moreover, modification of several mitochondrial proteins by succinylation is suggested to impose dysfunctional consequences [29, 30]. Thus, dysfunctional brain mitochondria reported in AD [31, 32] may account for the reduction of some DCAs, while oxidative damage of brain lipids resulting in the loss of brain tissue in AD [33] would increase urinary excretion of oxidized DCAs [34].

It has recently been recognized that some DCAs have receptors that control signaling and immune functions. Thus, changes in DCA biosynthesis may alter critical pathways resulting in abnormal AD brain function. Recent studies report higher longer-chain DCA plasma levels in stroke patients [35], further validating a role for these metabolites in neurovascular processes linked to brain disease.

In our current studies, we examined urine from individuals that were selected at higher risk for AD because of their age, and classified them as CH after an extensive neuropsychometric battery and the Uniform Data Set-2 criteria of the National Alzheimer's Coordinating Centers (NACC) [36]. We previously reported the logistic regression model based on CSF amyloid and Tau levels that correctly classified individuals with clinically probable AD [37] and used this to distinguish age-matched CH individuals with normal amyloid/Tau (CH-NAT) or pathological amyloid/Tau (CH-PAT). In a four year follow up, none of the CH-NATs but 40% of the CH-PATs declined cognitively [38]. Our hypothesis is that oxidative stress damages essential brain lipids and results in abnormal mitochondrial function that may contribute to neuronal dysfunction in early AD pathology. Our data supports this hypothesis by showing a group of shorter chain urine DCAs (C4, C5) that are lower in AD while another group of medium chain DCAs (C7, C8, C9) that are higher in AD urine compared to CH. We propose that DCA assays of urine, the epitome of an accessible sample for non-invasive AD screening, may offer a quantitative measure of both the increase in lipid oxidation and the reduction in energy components in AD.

## Materials and methods

### Ethics statement and diagnosis of study participants

The clinical research review committee of Huntington Medical Research Institutes and the Institutional Review Board of Quorum, protocol 33797, approved the protocol and consent forms for this study. All study participants gave written, informed consent. Primary caregivers who had durable power of attorney for health for participants who had diminished capacity because of diagnosis of clinically probable AD also approved and signed the consents. Participants between 70 and 100 years of age were recruited from the greater Los Angeles area, and medical and neuropsychological diagnostic processes for this study have been previously described [37]. Initially, the study participants were divided based on neuropsychological studies into 2 groups, cognitively healthy (CH, n = 76) and clinically probable AD (AD, n = 25). Based on our published method, we further divided the CH group into asymptomatic low risk individuals (CH-NAT, n = 45), and asymptomatic high risk individuals (CH-PAT, n = 31), based on beta amyloid$_{42}$/Tau ratios in the cerebrospinal fluid (CSF) [37].

### Measures of brain volume by MRI

The MR datasets were obtained using a GE 3 or 1.5T MR scanner with a standard eight-channel array head coil at HMRI. Anatomical sagittal spin echo T2-weighted scans were first obtained through the hippocampi (TR/TE 1550/97.15 ms, NEX = 1, slice thickness 5 mm with no gap, FOV = 188 x 180 mm, matrix size = 384 x 384). Baseline sagittal T1-weighted maps were then acquired using a T1-weighted 3D fast spoiled gradient echo (FSPGR) pulse sequence and variable flip angle method using flip angles of 2˚, 5˚ and 10˚. Data was analyzed using Freesurfer 6.0 (Freesurfer, Harvard) to obtain hippocampal and occipital lobe volumes (in μL).

### Urine collection, total protein, albumin, and creatinine

A single point mid-stream specimen of urine was collected from study participants after an overnight fast, between 8:00 am and 10:00 am. After centrifugation to remove any debris, urine was fractionated and stored in polycarbonate tubes at -80˚C until required for analyses. Urine was diluted (10-20X) and levels of creatinine determined using the improved Jaffe method using picrate using creatinine (0–15 mg/dL) as a standard (Creatinine kit, # 500701, Cayman Chemical Company, Ann Arbor, MI). Urine albumin was quantified using size

exclusion chromatography (HP1050) on a Zorbax GF-250 column (4.6 x 250 mm) using 0.1 PBS (pH 7.0) at a flow rate of 0.5 mL/min. The column was calibrated with thyroglobulin (670 kDa), gamma globulin (158 kDa), ovalbumin (44 kDa), myoglobulin (17 kDa), and vitamin B-12 (1.35 kDa) and levels of albumin calculated (mg/mL).

## Materials

HPLC grade water, ethyl acetate, and derivatizing reagents pentafluorobenzyl bromide (PFBBr) and anhydrous acetonitrile were purchased from Fisher Scientific. Hydrochloric acid, sodium sulfate, sodium chloride, non-deuterated dicarboxylic acids, dodecane, and N, N-dii-sopropylethylamine (DIPEA) were purchased from Sigma-Aldrich. Deuterated DCA standards (succinic acid-$d_4$, adipic acid-$d_4$, suberic acid-$d_4$, sebacic acid-$d_{16}$) were purchased from Cambridge Isotope Laboratory.

## Dicarboxylic acid extraction and derivatization

The extraction protocol was adapted from Costa *et al* [39]. Briefly, 500 μL urine and 100 μL deuterated internal standard mixture at 20ng/μL in ethanol was diluted to 1 mL with brine solution and acidified to pH 2 with 3 drops of 1 M HCl. Then, the urine was extracted 3 times with 3 mL ethyl acetate. The combined organic layer was dried with sodium sulfate before decanting and drying under a stream of nitrogen at 45˚C. Once dry, the extracted DCA were converted to dipentafluorobenzyl esters by adding 25 μL of 5% v/v PFBBr and 25 μL 10% v/v DIPEA in anhydrous acetonitrile to the residue. The reaction was allowed to proceed for 30 min at 60˚C. The reaction solution was then dried under a stream of nitrogen before adding 1 mL of hexanes to the reaction tube, vortexed for 10 min, and then transferred to GC/MS vials. After evaporation under a stream of $N_2$, the derivatized residue was dissolved in 100 μL dode-cane for GC/MS analysis.

## GC-MS analyses of derivatized dicarboxylic acids

DCA have two reactive carboxylic acid groups, making the parent mass M+2PFB. [M+1PFB]⁻ carboxylate ions (m/z) were detected by injecting 1 μL derivatized extracts onto a 7890A GC system coupled to a 7000 MS Triple Quad (Agilent Technologies). Gas chromatography was performed over 21.2 min using a Phenomenex Zebron ZB-1MS capillary GC column (2x15 m length, 0.25 mm I.D., 0.50 μm film thickness) heated to 150˚C for 1.2 min, ramped to 270˚C at 20˚C/min, and held for 2 min, then ramped to 340˚C at 10˚C/min and held for 5 min. The temperature of the ion source was 200˚C and the temperature of the quadrupoles was 150˚C. Single ion monitoring (SIM) was used to measure the [M+1PFB]⁻ carboxylate ions after nega-tive ion chemical ionization using methane gas.

The reproducibility measures (SD) when repeating the entire preparation and GCMS of the same original sample was < 20%; the SD when running the same sample by GCMS on conse-cutive days was < 6%. The list of carboxylate ions (m/z) for non-deuterated and deuterated dicarboxylic acid standards, retention times, linear ranges, and limits of detection are shown in S3 Table. The total ion chromatogram obtained from the GC/MS and the structures of the C3-C10, including the ionized PFB product are in the S1 Fig. along with the overall GCMS method.

## Data and statistical analyses

Agilent MassHunter Workstation Software was used to analyze GC/MS data. A calibration curve was acquired prior to sample analysis and quality control standards were analyzed after

every 10 samples. All samples were analyzed in triplicate. Peak integration was automatic for most fatty acids and manual integration was used in selected cases when automatic integration failed. We examined the mass of DCA normalized to volume, and then determined the percent distribution and proportion of the DCAs. By utilizing the percentage, we are able to reduce the coefficient of variation and also account for hydration as the percentages represent how each species relates to each other. Mann Whitney U tests were used to test for differences in DCA levels between cognitive groups. General linear models with Tukey-Kramer pairwise comparisons were used for multivariable analysis of C4-C5 and C7-C9 DCAs to adjust for potential confounders. Candidate cofounders were age, sex, smoking status, and Stroop Interference score, well-known predictors of cognitive health. Smoking status was modeled as a continuous variable (0 = never, 1 = ex, 2 = current). Multinomial logistic regression was used to predict cognitive group based on DCA levels.

All data analyses were performed using GraphPad Prism software or SAS v9.4 (SAS Institute, Inc., Cary, NC) and 0.05 was used as the significance level for all tests. Additionally, we used Spearman's rank correlation to evaluate relationships between DCA species and CSF levels of Ab and Tau proteins and also selected brain volumes as determined by MRI.

## Results

We recruited 100 study participants > 70 years of age and classified individuals by NACC UDS-2 criteria and consensus conferencing as cognitively healthy (CH, n = 76) or probable AD (n = 25). Those with mild cognitive impairment were excluded to reduce heterogeneity in the analysis. CH individuals were sub-classified by CSF $A\beta_{42}$ and Tau into CH-NAT (n = 45), or CH-PAT (n = 31). The groups were of similar age, and women comprised 58.3–66.7% across the groups (Table 1). We genotyped these individuals to determine their ApoE status, and compared their BMI and average number of years of education. In the latter case, AD individuals had less formal education than CH (p = 0.036), typical for AD [40].

To account for kidney function and hydration levels [41–43], we analyzed the urine concentration of total protein, creatinine, albumin, and the urinary albumin to creatinine ratio (UACR). Individuals with AD showed evidence of kidney function impairment through higher mean concentrations of total protein, albumin, and UACR compared to controls (Table 1), consistent with the higher level of albuminuria recognized with cognitive decline [44–46].

### Detection of dicarboxylic acids in urine

We quantified 8 DCAs in urine from cognitively healthy and AD individuals: malonic (C3), succinic (C4), glutaric (C5), adipic (C6), pimelic (C7), suberic (C8), azelaic (C9), and sebacic acids (C10). C4 accounted for with the majority of (42%, range 34.7%– 44.1%) of DCAs detected in urine while C6—C9 each represented >10% of total urine DCA (Fig 1A). C5, C3, and C10 accounted 6%, 3% and 2% of total urine DCA, respectively.

### Urine dicarboxylic acid species differ in CH compared with AD

The mean (+ standard deviation) total amount of DCA species in urine was 6.68 ± 3.92 μg/mL and 7.86 ± 4.54 μg/mL for CH and AD clinical groups, respectively. Urine DCA levels before and after logarithmic normalization are shown in Supplementary S1 Table & S2 Table. While there was no significant difference between the total amount of DCA species, for some individual acids mean levels were significantly higher in the AD group compared to the CH group (Fig 1B): pimelic, p = 0.0033; suberic, p = 0.0175; azelaic, p = 0.0010; and sebacic acids, p = 0.0051. To normalize between urine samples, levels of individual DCA species were

**Table 1. Demographic, clinical, and CSF/urine biomarkers.**

| | Clinical Classification | AD | All CH | CH | CH |
|---|---|---|---|---|---|
| | | (n = 25) | (n = 76a) | | |
| | CSF Aβ$_{42}$/Tau Classification | | | NAT | PAT |
| | | | | (n = 45a) | (n = 31) |
| | Mean Age ± SD (Range) | 79.2 ± 7.31 (62–91) | 78.0 ± 6.45 (63–91) | 77.3 ± 6.79 (63–90) | 79.1 ± 5.88 (68–91) |
| | % Female | 58.3% | 65.8% | 66.7% | 64.5% |
| | Smoking (0, past 1; present 2) | 0.64 ± 0.49 | 0.54 ± 0.53 | 0.45 ± 0.5 | 0.66 ± 0.54 |
| | ApoE Genotype | | | | |
| | 2/2 | 0 | 0 | 0 | 0 |
| | 2/3 | 0 | 18.3% | 17.1% | 20% |
| | 2/4 | 0 | 2.8% | 0 | 6.7% |
| | 3/3 | 66.7% | 57.7% | 68.3% | 43.3% |
| | 3/4 | 33.3% | 21.1% | 14.6% | 30% |
| | 4/4 | 0 | 0 | 0 | 0 |
| | BMI | 25.45 ± 4.92 | 26.63 ± 5.03 | 26.81 ± 5.55 | 26.38 ± 4.24 |
| | Education in Years | 14.75 ± 2.71 | **16.55 ± 2.53**[**] | **16.51 ± 2.39**[**] | **16.61 ± 2.75**[*] |
| CSF | Aβ$_{42}$ ± SD (95% CI) [pg/mL] | 536.9 ± 236.5 (437.0–636.8) | **759.1 ± 306.5**[**] (689.0–829.1) | **915.4 ± 247.6**[***] (841.0–989.8) | 532.1 ± 234.6 (446.0–618.1) |
| | Total Tau ± SD (range) [pg/mL] | 417.1 ± 169.9 (345.3–488.8) | **261.2 ± 148.5**[***] (227.3–295.2) | **187.1 ± 71.05**[***] (165.7–208.4) | 368.9 ± 165.8 (308.0–429.7) |
| Urine | Total Protein ± SD (95% CI) [μg/mL] | 182.7 ± 95.8 (142.3–223.2) | **136.9 ± 72.62**[*] (120.3–153.5) | **135.6 ± 78.27**[*] (112.1–159.1) | 138.9 ± 64.75 (115.1–162.6) |
| | Creatinine ± SD (95% CI) [μg/mL] | 1218.0 ± 720.4 (913.4–1522) | 1025.9 ± 550.5 (900.1–1152) | 1019.4 ± 558.3 (851.6–1187) | 1035.4 ± 548.0 (834.5–1236) |
| | Albumin ± SD (95% CI) [μg/mL] | 37.80 ± 25.44 (27.06–48.55) | **25.97 ± 31.86**[**] (18.64–33.30) | **24.25 ± 23.21**[**] (17.19–31.30) | **28.42 ± 41.49**[*] (13.20–43.64) |
| | UACR ± SD (95% CI) [mg/g] | 34.75 ± 23.21 (24.95–44.55) | **28.66 ± 38.51**[*] (19.80–37.52) | **29.34 ± 45.06**[**] (15.64–43.04) | 27.69 ± 27.30 (17.68–37.71) |

(a), for measures involving albumin, one value for CH is missing, so the n decreases by 1 in CH and CH-NAT

[*]p < 0.05

[**]p < 0.01

[***]p < 0.0001 versus AD.

expressed as a percentage of total DCA species. Mean proportions of succinic (p = 0.0113) and glutaric acids (p = 0.0087) were significantly lower in AD compared to CH. On the other hand, mean proportions of pimelic (p = 0.0035), suberic (p = 0.0161), and azelaic acids (p = 0.0022) were significantly higher in AD compared to CH (Fig 2A). The accuracy of the clinical group classification was enhanced when we combined the sum of metabolic process DCAs and the sum of oxidized products of longer chain fatty acids, as illustrated by lower p values (sum of C4 and C5: p = 0.0059; sum of C7 through C9: p = 0.0004), Fig 2B.

When we further sub-classified the CH group based on CSF amyloid and total Tau levels to distinguish those CH individuals at higher risk of developing AD, we identified three groups using pattern of CH-NAT, CH-PAT, and AD. Examination showed that the DCA group that was higher in AD is mainly derived from the breakdown of unsaturated fatty acids while the DCA group that was lower in AD is composed of components of the TCA cycle. Therefore, we combined these DCA groups separately and evaluated differences among CH-NAT, CH-PAT and AD (Fig 3).

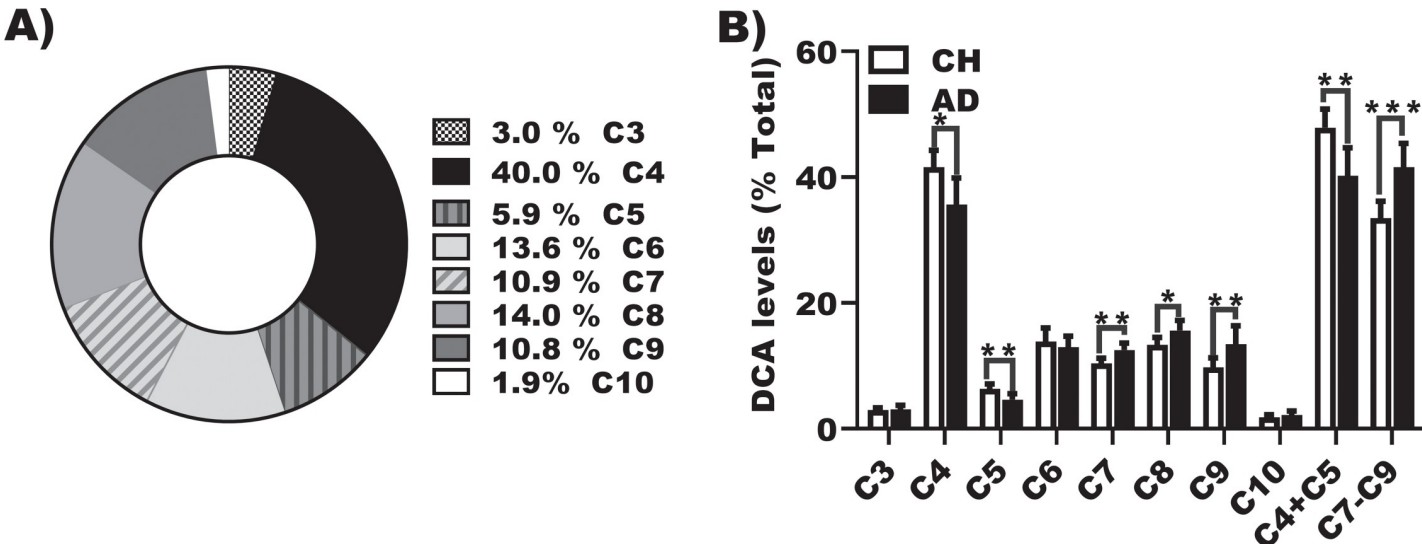

**Fig 1. Distribution and changes in urine DCA in clinical groups.** A) Total DCA was determined for all clinical groups, and we calculated the levels of each DCA as a percentage of the total of all species measured in urine. B) DCA intergroup comparison of each species (C3-C10), and of the sum of the major species that decrease (C4 +C5), and species that increase (C7+C8+C9) for CH (n = 76) compared with levels in AD (n = 25). *p < 0.05, **p < 0.01, and ***p < 0.001.

## Multivariable analysis of urinary DCA changes for C4/C5 and adjustment for multiplicity

Of the candidate confounders age, sex, smoking status, and Stroop Interference score, only smoking status was close to being a significant independent predictor of C4/C5 (p = 0.07). With smoking status included as a covariate and using the Tukey-Kramer adjustment for multiplicity, there was a significant difference between CH-NATs and CH-PATs (p = 0.04), and between CH-NATs and AD (p = 0.0004), but not between CH-PATs and AD (p = 0.26).

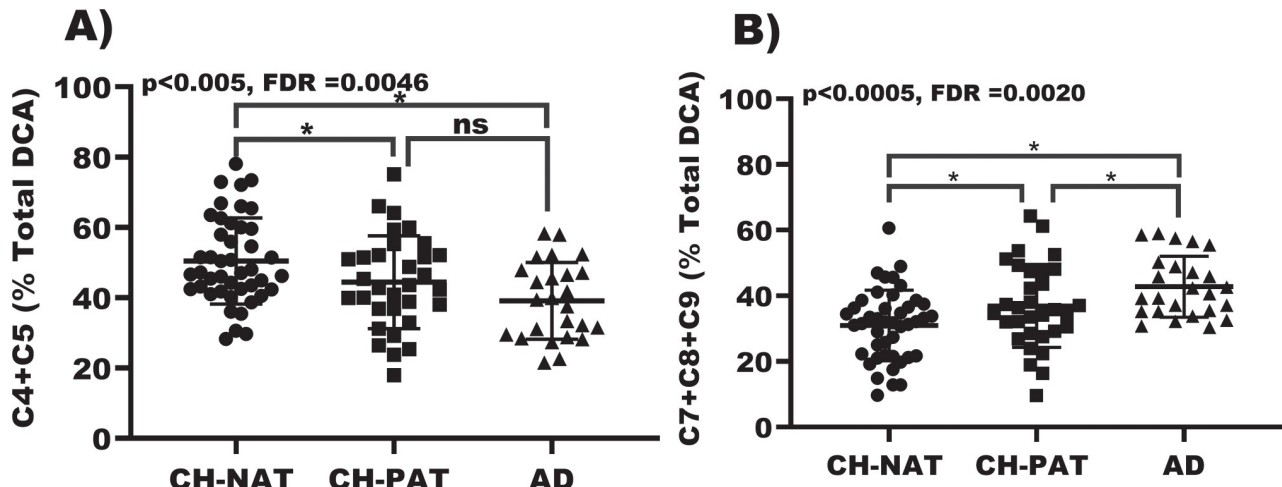

**Fig 2. Changes in urine DCA in clinical groups.** A) Violin plot and 1-way Anova with Fishers's LSD of C4+C5 in CH-NAT, CH-PAT, and AD. B) Violin plot and 1-way Anova with Fishers's LSD of C7+C8+C9 in CH-NAT, CH-PAT, and AD. *p < 0.05; ns not significant.

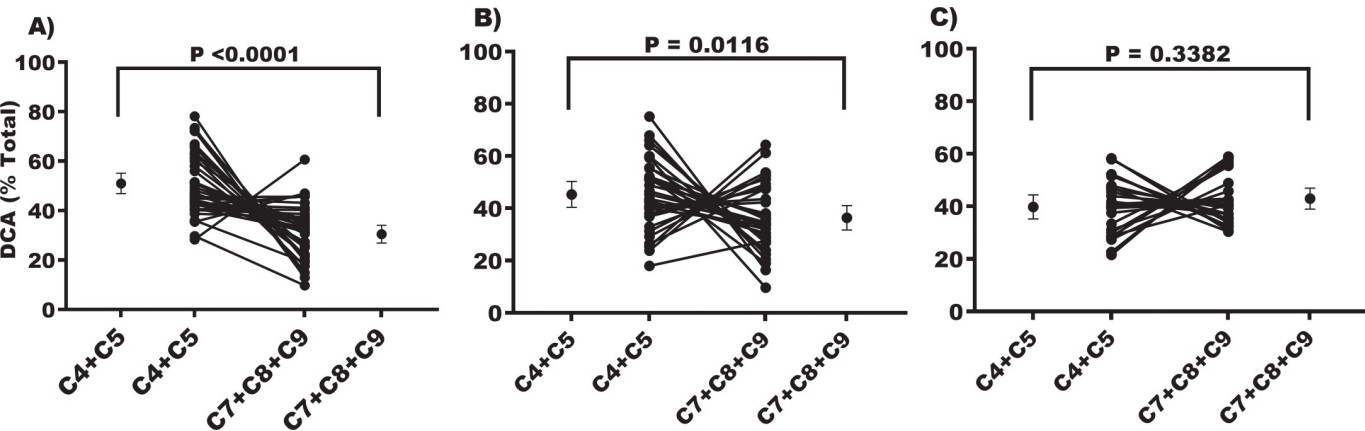

**Fig 3. Changes in energy versus oxidative DCA species in clinical groups.** A) Individual line graphs and the means (± 95% CI) of C4+C5 and C7+C8+C9 in urine from CH-NAT study participants. B) Separate line graphs and the mean (± 95% CI) of C4+C5 and C7+C8+C9 in urine from CH-PAT study participants. C) Individual line graphs and the means (± 95% CI) of C4+C5 and C7+C8+C9 in urine from CH-NAT study participants.

### Multivariable analysis of urinary DCA changes for C7-C9 and adjustment for multiplicity

For C7-C9, only age was close to being a significant independent predictor (p = .10). With age included as a covariate and using the Tukey-Kramer adjustment for multiplicity, the comparison between CH-NATs and AD was highly significant (p = 0.0002) whereas the comparisons between CH-PATs and CH-NATs and between CH-PATs and AD were not significant (p = 0.09 and 0.12, respectively).

### Predictive ability of DCAs for clinical and CSF classification

We tested how well a multinomial logistic model predicted membership to CH-NAT, CH-PAT, and AD groups based on C7-C9 DCAs. The model correctly predicted group for 46 of 101 (45.5%) individuals based on their C7_C9 values: 36 of 44 CH-NAT (82%) but only 2 of 32 CH-PAT (6%) and 8 of 25 AD (32%). Specificity for CH-NAT, CH-PAT, and AD was 42% (24/57), 86% (59/69), and 84% (64/76), respectively.

### Urine DCAs correlated with CSF and MRI biomarkers of AD

To determine if urinary DCA species may relate to brain degeneration, we looked at correlations between DCA and CSF $A\beta_{42}$ and Tau protein levels. The scatter plots (Fig 4) show that glutaric acid positively correlated with $A\beta_{42}$ (r = 0.23; p = 0.0186) while azelaic acid negatively correlated with $A\beta_{42}$ (r = -0.26; p = 0.0101). We found positive correlations with CSF Tau for azelaic (r = 0.22, p = 0.0276) and sebacic acids (r = 0.20; p = 0.0476) individually, and for the sum of C7-C10 (r = 0.20; p = 0.0499).

We tested whether the breakdown species C7 through C10 could be linked to the hippocampal volume by magnetic resonance imaging (MRI). Fig 5 shows a negative correlation between the percentage of breakdown species and hippocampal volume (left: r = -0.47; p = 0.0056, right: r = -0.49; p = 0.0040, total: r = -0.48; p = 0.0041, A-C). In contrast, we found no correlation between the combined C7-10 DCAs with the lateral occipital lobe volume, selected as a control region that is marginally affected in Alzheimer's disease.

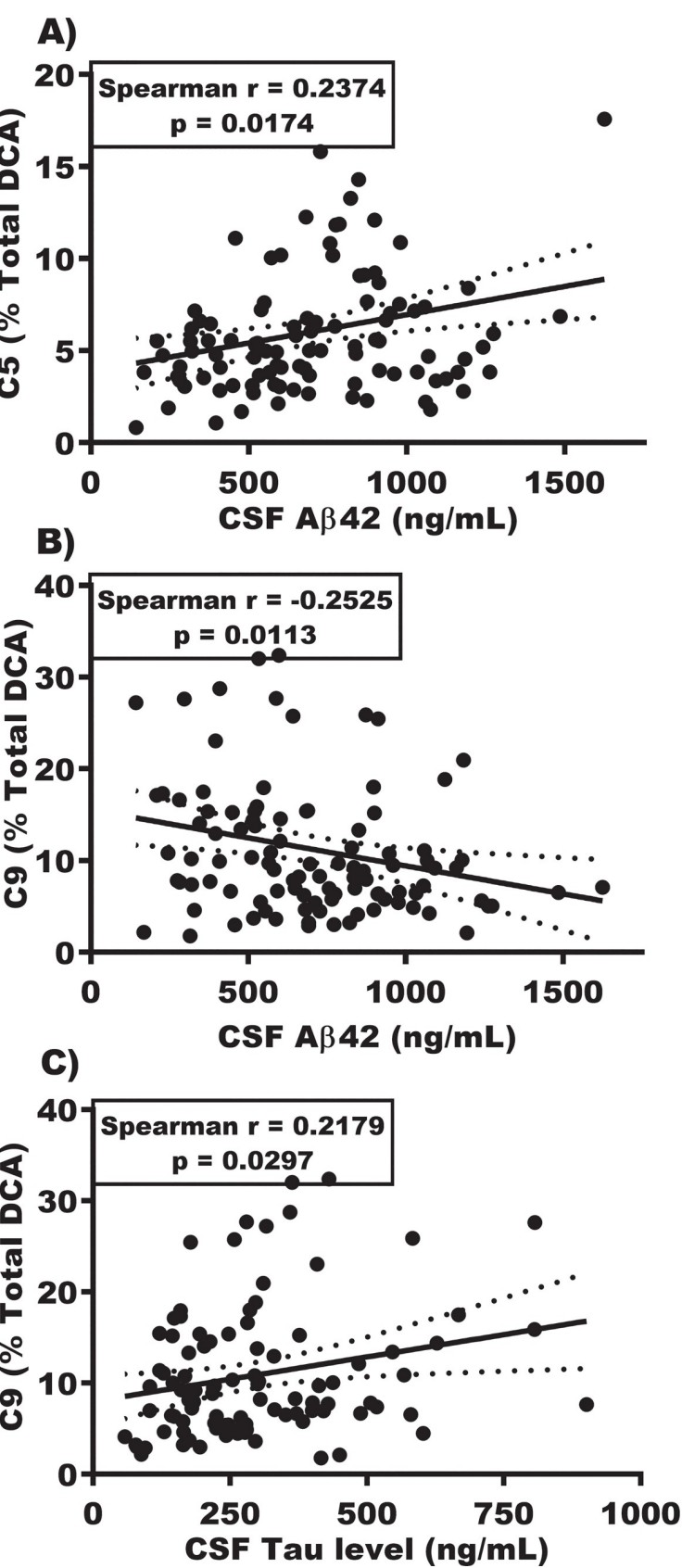

Fig 4. Correlation of urine DCA with CSF markers. A) Correlation of the proportion of (% Total) of C5 (glutaric acid) with CSF $A\beta_{42}$. B) Correlation of the percentage of C9 (azelaic acid) with CSF $A\beta_{42}$. C) Correlation of the percent C9 (azelaic acid) with CSF total Tau.

## Discussion

A non-invasive test with wide applicability is needed for early screening and diagnosis of AD pathology. Our strategy to analyze specific molecular components in urine stems from the major loss of brain volume and the decrease in energy capacity that are characteristic of AD. Females lose 9% and males 5.6% of brain tissue mass during AD progression [33]; losses of this degree from a brain of over 1.2 Kg should be noticeable, but the tissue loss occurs over many years. The excess metabolites have to leave the body and urine is the major mode of excretion.

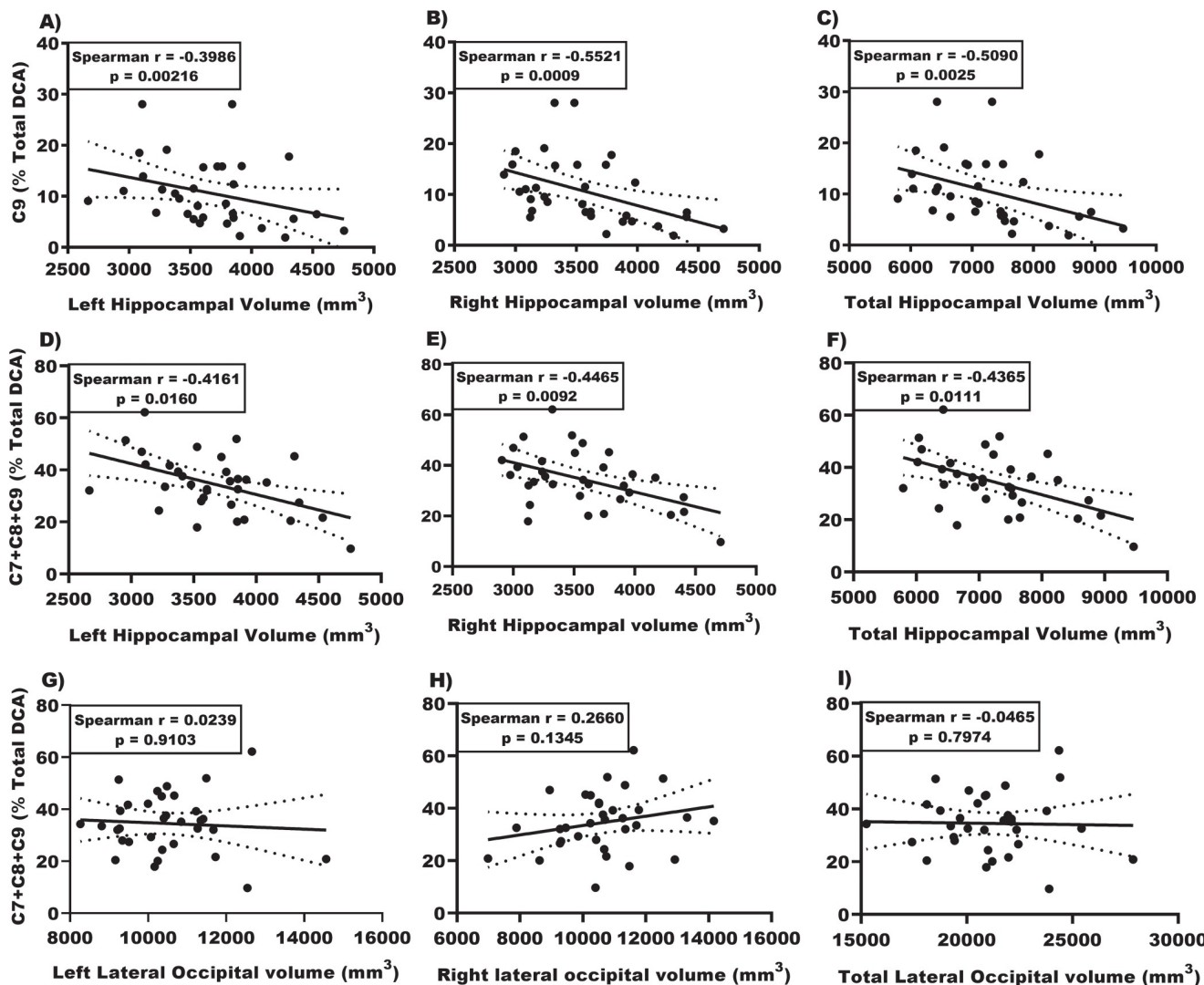

Fig 5. Correlation of urine DCA levels with brain volumes. Top graphs. Correlations of C9 levels in urine with left (A), right (B), and total hippocampal (C) volume for CH individuals. **Middle graphs: Correlations of the sum of DCAs (C7+C8+C9) that increase in urine with left (D), right (E), and total hippocampal volumes for CH individuals (F).** Bottom graphs: Correlations of the sum of DCAs (C7+C8+C9) that increase in urine with left (G), right (H), and total lateral occipital volumes (I) for CH individuals.

However, several challenges exist and must be addressed. Could the long duration of neurodegeneration and the variability of urine production, coupled with contributions to urine from the entire body, make it difficult to assign any such change to AD pathology? Our hypothesis is that the predominant neurodegenerative process is unique to brain, such that known brain-enriched lipids might be identified in the urine, and their changing amounts will indicate their origin by their correlation with CSF and brain biomarkers of AD. Another consideration is which lipids to choose from the complex mixtures that may originate from brain tissues. For example, Montine and colleagues reported increased lipid oxidation in CSF from AD patients by measuring oxidative products [47], but the changes in urine F2-isoprostanes were not reliable [47]. We chose to examine urine excretion of DCAs to test if known features of AD pathology (energy metabolism and tissue integrity) can be recognized. These DCAs have not been quantified previously in urine from AD patients and are end-products of β- or omega-oxidation on unsaturated fatty acids (UFAs) [48]. Higher levels of oxidized DCAs are detected in urine of rodents and humans with metabolic syndromes or pathological conditions, or when treated with xenobiotic drugs [49, 50].

We first studied the urine levels of glutaric and succinic acids that are integral to the Krebs cycle, since energy loss and mitochondrial dysfunction are established in AD pathology [31, 32]. The lower levels of both of these DCAs in urine that we observed in AD compared to CH support our hypothesis. The combined C4 and C5 levels were also numerically lower in CH-PATs than in CH-NATs; though not significant, the distribution was convincingly midway between CH-NAT and AD levels. The significant positive correlation between glutaric acid (C5) levels in urine and CSF Aβ$_{42}$ levels supports our hypothesis that the reduction in glutaric acid reflects a loss of energy capacity resulting from AD neurodegeneration at least partially and, importantly, this change starts in the pre-symptomatic CH-PAT phase when individuals are clinically indistinguishable from CH-NATs. These data suggest succinic and glutaric acid levels in urine offer a potential screen for the reduction in brain energy capacity in early AD pathology.

We next tested if the oxidized DCAs [51] [adipic (C6), pimelic (C7), suberic (C8), azelaic (C9), and sebacic acids (C10)] might represent loss of brain tissue by comparing their levels with CSF and brain MRI biomarkers if AD. We hypothesized that the vulnerable double bonds in the brain-enriched UFAs would be increasingly oxidized and their break down products excreted in greater amounts in urine as the AD pathology and its tissue loss progress. Our findings suggest that C6 is not altered but levels of C7 through 10, individually and combined in C7-10, are increased in AD compared to CH. Especially interesting was that the combined measure of C7-9 DCAs was increased in CH-PATs compared to CH-NATs, mainly from the rise in azelaic acid (C9 DCA). The higher level of C9 also weakly correlates with the AD CSF biomarkers: negatively with Aβ$_{42}$ and positively with CSF Tau. An independent measure of AD, a lower hippocampal volume, also correlated with the higher urine levels of the oxidized C7-9 panel. This combined match between C7-9 and the AD biomarkers in CSF and brain MRI support the hypothesis that the C7-9 excretion is linked in part to the neurodegenerative loss of brain in AD.

## Limitations

While our study has significant implications for AD research, limitations of this exploratory report are worth discussion. As an exploratory study, there were no *a priori* group differences for which specific hypothesis testing was done because we have no knowledge on differences that would be clinically meaningful; our study is a hypothesis-generating, first step in addressing that. Distinguishing AD from physiological variations in urine lipids is a challenge, since

variation in urine composition can arise from many factors, in particular diet, hydration, time of collection, and kidney disease, for which a number of correction approaches have been used [52]. Further, microalbuminuria is known to occur with cognitive decline [44–46]. We chose to collect "spot" urine samples rather than the 24-hour collections that average changes throughout the day, because we intend that any future test in clinical practice must implement a simple collection process. To reduce effects from physiological variables in our study, we collected urine "spot" samples after a 12 hour overnight fast, excluded the one individual that had proteinuria (> 300 μg/mL), and normalized individual DCA levels to the total amount of DCAs measured, in units of percent. Specificity of clinical or biochemical correlation will require replication by other labs, testing in different populations, longitudinal studies, and matching with other biomarkers and clinical outcomes, including brain pathology. Finally, the complex sample preparation and GCMS methods have more variability than simpler assays such as ELISAs. These initial results of C3-10 DCAs may encourage further research of additional urine lipid species. Our methods section reports our reproducibility within different preparations of the same sample, as well as for samples taken on consecutive days from the same individuals. These current methods are not conducive, however, to widespread adoption or for good inter-laboratory reproducibility, and simplified methods are needed; this is a technical issue that can be overcome now that a pathological marker has been identified.

## Energy and amyloid removal

Amyloid change remains a cornerstone of AD pathology and reasons for its accumulation in AD brain are widely explored. Studies by Fila and colleagues suggest that immune cells utilize omega-3 fatty acids for energy generation and enhanced removal of amyloid [53]. Moreover, omega-3 fatty acids also stimulate signaling pathways [54]. Given the decrease in omega-3 fatty acids in CSF in our previous study [55] concomitant with the increase of their oxidative breakdown products in urine of AD study participants in the present study, we propose that immune processes that enhance amyloid removal, cell signaling pathways that enhance neuronal function, and the basic energy requirements of the brain may be compromised in AD.

## Signaling and brain function

Recent studies have shown that lysine succinylation is a post translational modification (PTM) that is evolutionarily conserved, is regulated by histone deacetylases [56] that are implicated in AD progression [57]. The importance of succinylation ranges from coupling metabolism with protein function in the nervous system [58, 59], association with energy regulation and several cellular metabolic processes including glucose metabolism [56, 60, 61]. Histone proteins that may directly regulate gene expression via chromatin reorganization are subject to acetylation and succinylation as a major post translational modification process [62]. Thus, alterations in succinate that we describe in our clinical groups suggest that these important succinylation pathways may be compromised in AD.

## Biochemical and clinical implications of the interaction of DCAs changes

Our studies show diametrically opposed changes in two groups of DCAs in urine (Fig 6). While energy-related C4/C5 are higher and oxidatively derived C7/C8/C9 are lower in cognitively healthy study participants, the opposite levels are present in the urine from AD participants. Functionally, these two groups of DCAs also have opposite effects. For example, succinate is a cofactor in energy metabolism via the TCA cycle while azelaic acid is known to inhibit several TCA enzymes and mitochondrial electron transport proteins. If the clearance of amyloid via autophagocytosis, the repair of post mitotic neurons, and other processes required

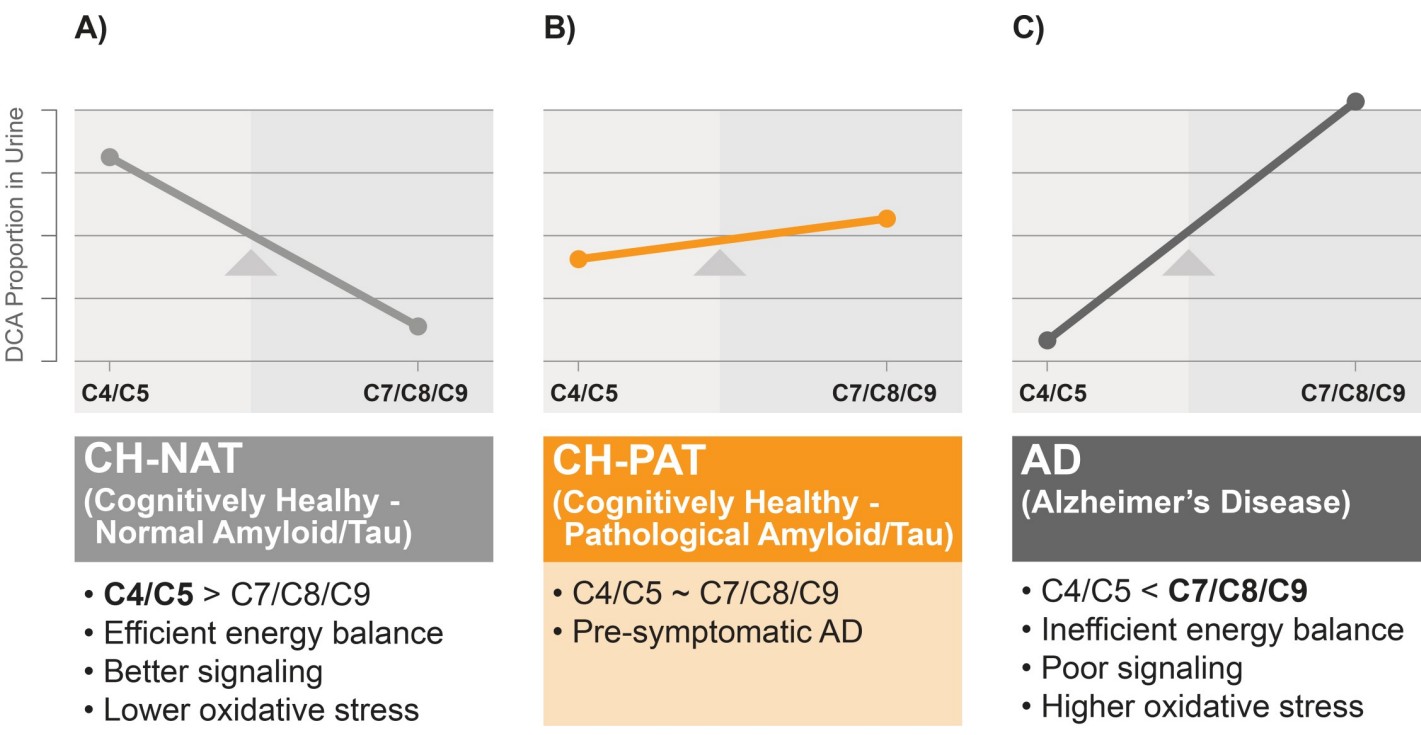

**Fig 6. Summary of major changes in urine DCA.** Our studies identify two groups of DCAs: the first group consisting of C4+C5 is linked with energy and signaling while the second group (C7+C8+C9) is derived from the oxidative breakdown of long-chain unsaturated fatty acids. A balance between energy production or signaling and oxidative breakdown classifies CH-NAT, CH-PAT, and AD. Energy producing DCAs are the dominant species in CH-NAT (A), there is equilibrium between these species in CH-PAT (B), and oxidatively-generated DCA species become prominent in AD (C). Measures that prevent the shift from C4/C5 to C7/C8/C9 may be useful in controlling AD progression.

for maintaining a healthy brain require energy, a higher C4/C5 and lower C7/C8/C9 is desirable. On the other hand, a lower C4/C5 and a higher C7/C8/C9 will favor the accumulation of amyloid, resulting in brain dysfunction that characterizes AD pathology. The implications of our study are that strategies that increase C4/C5 and decrease C7/C8/C9 can enhance cognitive function or diminish AD progression. Evidence in support of this comes from studies showing that dietary supplementation with TCA components can prevent cognitive decline [63] and prevent amyloid toxicity [64]. Examination of strategies that regulate these two groups of DCAs is now warranted. Determining the ratios of these two DCA classes may also be a useful diagnostic tool.

Overall, we justify an aggressive pursuit of this approach, since our data provide multiple lines of support for the hypothesis that urine DCAs mirror the neurodegeneration of early pathology and reflect two different pathophysiological processes of AD: loss of energy and brain tissue. Urine C4 and C5, and C7-9 are promising tools to quantify a decrease in brain energy capacity and/or loss of brain tissue respectively. These new urine biomarkers may help screen individuals as we age and suggest urine may hold value as a source of insight to health versus early AD.

## Supporting information

**S1 Table. Distribution, proportion, and intergroup comparisons of DCA species normalized for urine volume (ng/mL) between clinical groups.** P values < 0.05 are shown in bold italics.
(DOCX)

**S2 Table. Percent distribution, proportion, and intergroup comparison of DCA species between clinical and biochemical groups.** P values < 0.05 are shown in bold italics.
(DOCX)

**S3 Table. Analytical parameters used for detecting and quantifying DCAs in urine samples.** Carbon number (C3-C10), negative ion (m/z), retention time (RT), deuterated internal standards, detection linear range, and correlation ($R^2$).
(DOCX)

**S1 Fig. Structures of the C3-C10 DCAs and GCMS methods.**
(DOCX)

# Acknowledgments

This project depended on the altruistic involvement of our study participants.

# Author Contributions

**Conceptualization:** A. N. Fonteh, M. G. Harrington.

**Data curation:** K. J. Castor, T. Tran.

**Formal analysis:** K. J. Castor, S. P. Edminster, T. Tran, K. S. King, J. M. Pogoda, A. N. Fonteh, M. G. Harrington.

**Funding acquisition:** M. G. Harrington.

**Investigation:** K. J. Castor, S. Shenoi, S. P. Edminster, T. Tran, K. S. King, J. M. Pogoda, A. N. Fonteh, M. G. Harrington.

**Methodology:** K. J. Castor, J. M. Pogoda, A. N. Fonteh, M. G. Harrington.

**Project administration:** M. G. Harrington.

**Supervision:** K. S. King, H. Chui, A. N. Fonteh, M. G. Harrington.

**Validation:** K. J. Castor, J. M. Pogoda.

**Writing – original draft:** A. N. Fonteh, M. G. Harrington.

**Writing – review & editing:** K. J. Castor, S. Shenoi, S. P. Edminster, T. Tran, K. S. King, H. Chui, J. M. Pogoda, A. N. Fonteh, M. G. Harrington.

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
