## [Decision Letter · Decision Letter 0]

5 Mar 2020

PONE-D-19-26326

Urine dicarboxylic acids change in pre-symptomatic Alzheimer’s disease and reflect loss of energy capacity and hippocampal volume

PLOS ONE

Dear Dr. Harrington,

Thank you for submitting your manuscript to PLOS ONE. After careful consideration by two Reviewers and an Academic Editor, please make the suggested corrections posed by both Reviewers so I can render a decision on this manuscript.

2.  Please describe in your methods section how capacity to provide consent was determined for the participants in this study. Please also state whether your ethics committee or IRB approved this consent procedure. If you did not assess capacity to consent please briefly outline why this was not necessary in this case.

"No. The funders had no role in study design, data collection and analysis, decision to publish, or preparation of the manuscript."

a)    Please provide an amended Funding Statement that declares *all* the funding or sources of support received during this specific study (whether external or internal to your organization) as detailed online in our guide for authors at http://journals.plos.org/plosone/s/submit-now.  

b)    Please state what role the funders took in the study.  If any authors received a salary from any of your funders, please state which authors and which funder. If the funders had no role, please state: "The funders had no role in study design, data collection and analysis, decision to publish, or preparation of the manuscript."

We note that one or more of the authors are employed by a commercial company: Cipher Biostatistics & Reporting.

**Comments to the Author**

1. Is the manuscript technically sound, and do the data support the conclusions?

Reviewer #1: Yes

Reviewer #2: Yes

2. Has the statistical analysis been performed appropriately and rigorously? 

Reviewer #1: Yes

Reviewer #2: Yes

3. Have the authors made all data underlying the findings in their manuscript fully available?

Reviewer #1: Yes

Reviewer #2: Yes

4. Is the manuscript presented in an intelligible fashion and written in standard English?

Reviewer #1: Yes

Reviewer #2: Yes

5. Review Comments to the Author

Reviewer #1: This is a very interesting paper. The reviewer agree with authors' hypothesis. But would like to suggest an open search for lipid metabolites as urinary biomarkers in the future. The manuscript should be published to show the community the potential of urine. But there is a theory about urinary biomarker which will support the manuscript but not mentioned. The theory is in the paper with recent progress of urinary protein biomarkers on early AD animal model (J Alzheimers Dis. (2018) 66:613–637. doi: 10.3233/JAD-180412).

Reviewer #2:

1. Please give the precise structure of C4-C5 DCAs and C7-C10 DCAs, are they oxidized?

2. Single ion monitoring (SIM) mode based on triple quad mass spectrometry is a Quantitative platform for targeted analysis, but is not suitable for the finding of biomarkers. Therefore, although the authors targeted dicarboxylic acids because they are implicated in several processes associated with AD pathology, I guess readers may also need to explain why and how the C4-C5 DCAs and C7-C10 DCAs were selected for SIM analysis. Specifically, whether some other DCAs such as smaller than C4 or bigger than C 10 were detected in urine, or there is no statistical differences for these DCAS between AD and cognitively healthy groups.

3. The authors have provided well designed and credible results, especially; the samples of CH individuals with normal amyloid/Tau (CH-NAT) and pathological amyloid/Tau (CH-PAT) are very valuable. However, some details of the analysis method may be ignored. For example, gas chromatographic methods coupled with mass spectrometry is widely used for the detection of dicarboxylic acids, but the efficiency and stability of reaction products and several derivatization agents may be different, so the detection limit (LOD) and relative standard deviation (RSD) of C4,C5… C10 should be given, 0therwise, the relationship between signal intensity and content cannot be accurately judged.

4. In Fig 2(B). Changes in urine DCA in clinical groups, if I am not wrong, there are 25 black triangles, which is not inconsistent with 24 AD samples.

6. PLOS authors have the option to publish the peer review history of their article (what does this mean?). If published, this will include your full peer review and any attached files.

**Do you want your identity to be public for this peer review?** For information about this choice, including consent withdrawal, please see our Privacy Policy.

Reviewer #1: No

Reviewer #2: Yes: Yong Wang

We would appreciate receiving your revised manuscript by June, 2020. To enhance the reproducibility of your results, we recommend that if applicable you deposit your laboratory protocols in protocols.io, where a protocol can be assigned its own identifier (DOI) such that it can be cited independently in the future. For instructions see: http://journals.plos.org/plosone/s/submission-guidelines#loc-laboratory-protocols

We look forward to receiving your revised manuscript.

Kind regards,

Stephen D. Ginsberg, Ph.D.

Section Editor

PLOS ONE

---

## [Author Response · Author response to Decision Letter 0]

26 Mar 2020

Responses to the reviewers

Reviewer 1

But would like to suggest an open search for lipid metabolites as urinary biomarkers in the future. We add a statement to recommend an open search for lipid metabolites in our “limitations” paragraph.

But there is a theory about urinary biomarker which will support the manuscript but not mentioned. The theory is in the paper with recent progress of urinary protein biomarkers on early AD animal model (J Alzheimers Dis. (2018) 66:613–637. doi: 10.3233/JAD-180412). We add the recommended reference.

Reviewer 2

1. Please give the precise structure of C4-C5 DCAs and C7-C10 DCAs, are they oxidized? We have added the structure of the DCA’s to the Supp figure 1. We have also included more GCMS method details with the structure of the PFBBr species that we detect in the method; this is interesting as only one PFBBr adduct is detected.

2. Single ion monitoring (SIM) mode based on triple quad mass spectrometry is a Quantitative platform for targeted analysis, but is not suitable for the finding of biomarkers. Therefore, although the authors targeted dicarboxylic acids because they are implicated in several processes associated with AD pathology, I guess readers may also need to explain why and how the C4-C5 DCAs and C7-C10 DCAs were selected for SIM analysis. Specifically, whether some other DCAs such as smaller than C4 or bigger than C 10 were detected in urine, or there is no statistical differences for these DCAS between AD and cognitively healthy groups. The recovery yield for C3 and DCAs greater than 10 turned out to be poor and we were not able to optimize the method within these experiments. The values for C6, and C10 were of good quality but they were not significantly different between the groups. We concentrated on those DCAs that we found to be different between CH and AD and which yielded good quality data. We have started to look at larger DCA’s with a new approach, but this is beyond the scope of this manuscript. We used a triple quad GCMS and used the SIM approach because the DCAs all fell apart under conditions we tried. As we mention in the limitations section, we state clearly that a simpler assay will be needed for widespread use, but we hope that these initial results that we report will stimulate efforts to look further at urine as a biomarker with potential for screening of pre-symptomatic AD.

3. However, some details of the analysis method may be ignored. For example, gas chromatographic methods coupled with mass spectrometry is widely used for the detection of dicarboxylic acids, but the efficiency and stability of reaction products and several derivatization agents may be different, so the detection limit (LOD) and relative standard deviation (RSD) of C4,C5… C10 should be given, 0therwise, the relationship between signal intensity and content cannot be accurately judged. In our original submission, we had included the LOD, dynamic ranges, 95% CI, means (SD) for all values of DCAs, and the R2 of internal standards in Tables 1-3. Perhaps the reviewer did not have access to this supplementary data; these parameters give clear description of the data quality.

4. In Fig 2(B). Changes in urine DCA in clinical groups, if I am not wrong, there are 25 black triangles, which is not inconsistent with 24 AD samples. Good catch! We had stated we had an N of 24 for AD in some texts, Tables and in scatter plots in figure 2B; it is 25! We had 25 correctly in other parts. We have corrected this mistake throughout, as indicated in the track changes. Thanks!

---

## [Editor Report · Decision Letter 1]

1 Apr 2020

Urine dicarboxylic acids change in pre-symptomatic Alzheimer’s disease and reflect loss of energy capacity and hippocampal volume

PONE-D-19-26326R1

Dear Dr. Harrington,

We are pleased to inform you that your manuscript has been judged scientifically suitable for publication and will be formally accepted for publication once it complies with all outstanding technical requirements.

With kind regards,

Stephen D. Ginsberg, Ph.D.

Section Editor

PLOS ONE

---

## [Editor Report · Acceptance letter]

6 Apr 2020

PONE-D-19-26326R1 

Urine dicarboxylic acids change in pre-symptomatic Alzheimer’s disease and reflect loss of energy capacity and hippocampal volume 

Dear Dr. Harrington:

I am pleased to inform you that your manuscript has been deemed suitable for publication in PLOS ONE. Congratulations! Your manuscript is now with our production department. 

With kind regards,

on behalf of

Dr. Stephen D. Ginsberg 

Section Editor

PLOS ONE